# A multi-reservoir extruder for time-resolved serial protein crystallography and compound screening at X-ray free-electron lasers

Maximilian Wranik [1,12] ✉, Michal W. Kepa [1,12] ✉, Emma V. Beale [2,12], Daniel James[1], Quentin Bertrand [1], Tobias Weinert [1], Antonia Furrer [1], Hannah Glover [1], Dardan Gashi[2], Melissa Carrillo [3], Yasushi Kondo[1], Robin T. Stipp[1], Georgii Khusainov[1], Karol Nass[4], Dmitry Ozerov [5], Claudio Cirelli [2], Philip J. M. Johnson [6], Florian Dworkowski [4], John H. Beale [4], Scott Stubbs[7], Thierry Zamofing[7], Marco Schneider[7], Kristina Krauskopf [8], Li Gao [8], Oliver Thorn-Seshold [8], Christoph Bostedt[2,9], Camila Bacellar [2], Michel O. Steinmetz [1,10], Christopher Milne [11] & Jörg Standfuss [1]

Serial crystallography at X-ray free-electron lasers (XFELs) permits the determination of radiation-damage free static as well as time-resolved protein structures at room temperature. Efficient sample delivery is a key factor for such experiments. Here, we describe a multi-reservoir, high viscosity extruder as a step towards automation of sample delivery at XFELs. Compared to a standard single extruder, sample exchange time was halved and the workload of users was greatly reduced. In-built temperature control of samples facilitated optimal extrusion and supported sample stability. After commissioning the device with lysozyme crystals, we collected time-resolved data using crystals of a membrane-bound, light-driven sodium pump. Static data were also collected from the soluble protein tubulin that was soaked with a series of small molecule drugs. Using these data, we identify low occupancy (as little as 30%) ligands using a minimal amount of data from a serial crystallography experiment, a result that could be exploited for structure-based drug design.

X-ray free-electron lasers (XFELs) have opened up a wealth of new structural biology research in recent years[1–4]. XFEL beams are several orders of magnitude brighter than third-generation synchrotron sources, and their high brilliance enables X-ray diffraction to be observed from even sub-micron-sized protein crystals[5,6]. The 'diffraction before destruction' method[7,8], achieved by the femtosecond pulse length in concert with the intense XFEL beam, allows diffraction data to be collected prior to the prevalence of radiation damage[9]. Consequently, experiments can routinely be performed at room temperature, offering high-resolution structural data that more closely represent proteins and ligands in their native environment[10]. The ability to work on non-cryogenic samples further facilitates time-resolved studies of protein dynamics and has resulted in a number of stop-motion 'movies' of biochemical processes[3,4] with temporal resolution down to the femtosecond range[11–13]. Despite such successes, XFEL beamtime availability is still limited and beamline operation costs are much higher compared to those of their synchrotron counterparts. Furthermore, the destructive properties of the XFEL pulses impose specific demands on sample preparation and delivery, namely, that a fresh sample must be delivered for each XFEL pulse – a method which

is referred to as serial crystallography[2]. Serial delivery to the XFEL beam has been achieved using a number of different methods including liquid jets[14], high-viscosity extrusion (HVE) injectors[15], fixed target supports[16], and acoustic droplet or 'drop-on-demand' systems[17]. Detailed comparisons of the advantages and disadvantages of these can be found in recent review articles[2,18–20].

The most appropriate sample delivery method is dictated by a number of parameters. These can include the properties and volume of sample available, the nature of the data that needs to be collected, the properties of the XFEL source, the compatibility of the sample delivery system with the experimental endstation, and the available support and expertise of the experimental team. Due to their sample exchange efficiency and compatibility with both soluble and membrane proteins, HVE injectors have become the most popular devices, now accounting for more than 40% of published serial femtosecond X-ray crystallography (SFX) studies[2]. At synchrotron sources, HVE injectors have also been adapted for both static[21–23] and time-resolved serial crystallography[24,25] for probing structural changes on the order of milliseconds. With the advent of diffraction limited 4th generation synchrotrons and beamlines either dedicated to or capable of serial crystallography measurements, it is evident that serial crystallography methods are becoming even more widespread. Examples currently in operation include MAX IV (BioMAX)[26], NSLS II (FMX)[27] and ESRF (ID29)[28]. Further examples of planned upgrades include the APS[29], SLS2.0[30], Diamond-II[31] and PETRA-IV[32]. These are expected to be operational in 2024, 2025, 2027 and 2029, respectively.

In this work, we describe a multi-reservoir device for viscous sample extrusion, which was designed to address some of the limitations of existing HVE injectors[15,23,33], particularly the timely manual exchange of sample reservoirs. The device was also designed to automate the sample exchange process in a serial crystallography experiment in general. The data collection efficiency was improved by minimising sample exchange time and user-beamline interaction. In addition, the device offers precise temperature control for stable sample storage but also for probing temperature-dependent processes and alteration of reaction kinetics. The applicability of the device for XFEL experiments is demonstrated in three steps. First, we install and validate the general performance of the device at the Alvra endstation of the Swiss X-ray free-electron laser (SwissFEL) using two viscous carrier media: the lipidic cubic phase of monoolein (LCP), which is used commonly for membrane protein crystallisation and hydroxyethylcellulose (HEC), a carrier matrix for soluble proteins. Second, we utilise the device in a time-resolved serial femtosecond crystallography (TR-SFX) experiment by collecting structural snapshots of the light-driven sodium pump *Krokinobacter eikastus rhodopsin 2* (KR2). Third, we demonstrate efficiency gains in ligand-soaking experiments for structure-based drug design (SBDD) approaches by solving the structure of the anticancer target tubulin in complex with both small molecules (some of which are approved chemotherapeutics) and with new photochemical affinity switches for the optical control of tubulin function. Based on these experiments, we demonstrate how ligands bound with low occupancy can be identified with a minimum amount of data and thus XFEL beamtime.

Efficient, precise and reliable sample delivery is indispensable in serial crystallography experiments, both when resolving the binding mode of drug molecules at room-temperature but perhaps even more so during time-resolved experiments. The latter often requires many datasets at different time-points in order to capture the intermediates of biochemical reactions. The multi-reservoir high viscosity extruder presented here constitutes an important step towards the automation and improved efficiency of serial crystallography data collection at XFELs and possibly 4th generation X-ray sources, which is likely to attract interest of new academic and industrial users.

## Results

### Features and design of the multi-reservoir high viscosity extruder

We designed and constructed the multi-reservoir extruder at the Paul Scherrer Institut (PSI), Switzerland. The device was installed in the Alvra Prime experimental chamber at the Alvra endstation of the Aramis beamline at SwissFEL[34,35]. It was fitted on top of the existing 3-axis stage in the Prime chamber, permitting alignment of the device with respect to the X-ray and optical laser beams. In all experiments, the chamber was operated in a low-pressure helium environment (100–800 mBar). The device schematics are shown in Fig. 1A with further details in Supplementary Figs. 1-4. The device comprises three key modules: the reservoir holder, the plunger assembly and the sample interaction region. Samples were housed in custom reservoir assemblies as shown in Fig. 1B and Supplementary Fig. 2. These key modules are described briefly below with further details of the multi-reservoir extruder presented in the **Supplementary Information**.

The reservoir holder is a revolving cylindrical drum containing nine slots. Each slot holds a single sample reservoir with a maximum volume of 130 µL, bringing the total sample capacity of the assembly to 1170 µL. Rotation of the reservoir holder drum via a stepper motor allows each of the nine sample reservoirs to be moved into position for sample injection. Temperature control of the reservoir is achieved using rotational liquid feedthrough channels that surround each slot (Supplementary Fig. 1A). The device offers temperature stability to within ±0.25 °C in the range of approximately 10 °C and 45 °C with an experimentally verified offset between the temperature set on the circulator and the measured temperature of the sample reservoir (Fig. 1C).

The sample interaction region is fixed by the geometry of the Alvra Prime chamber. Reservoirs are driven into the interaction region in order to extrude the sample into the focused X-ray and optical laser pathways. The loading sequence is as follows: once the desired reservoir has been rotated in position, a stepper motor translates the entire reservoir holder and plunger assembly together towards the sample interaction region. This translation is controlled automatically based on positional feedback from an LED-light curtain. A guiding aperture then ensures an accurate fit of the capillary needle of the sample reservoir into the ceramic gas aperture. The gas aperture is connected to a helium gas supply fitted with a remotely controlled inline pressure regulator. Additionally, downstream of the sample jet, a suction nozzle (catcher), as well as a capillary tip cleaner are installed (Supplementary Fig. 3). The former helps stabilising sample jet formation; the latter allows for on-the-fly cleaning of the capillary. Both cleaning and exchange of sample can be done without breaking the Helium environment of the Prime chamber, increasing the experiment's efficiency by minimizing latent machine time. Supplementary Movie 1 illustrates the sample exchange process using the multi-reservoir extruder.

The plunger assembly is responsible for the extrusion of the sample into the path of the X-ray beam through the capillary needle. Unlike most designs[15,23,33], where a high-performance liquid chromatography (HPLC) pump is used to extrude the sample, our device utilises force from a mechanical motor. Positive and negative limit switches permit a precise read out of how much sample volume has been loaded and is remaining at any given point during extrusion. A load cell allows for monitoring of the pressure exerted on the plunger and thus a relative measure of the pressure on the sample during extrusion (Supplementary Fig. 4). This pressure reading provides feedback on sample homogeneity, jet stability, and capillary blockages so that suboptimal data due to inconsistent jetting speeds can be excluded from downstream processing. Furthermore, since sample volumes are sometimes less than the maximal 130 µL permitted by the reservoirs, the load cell allows for 'fast' motion of the plunger up until it hits the sample (at which point a pressure spike is used to immediately stop the motion), which significantly reduces the initial time taken for the plunger to reach the sample. During the

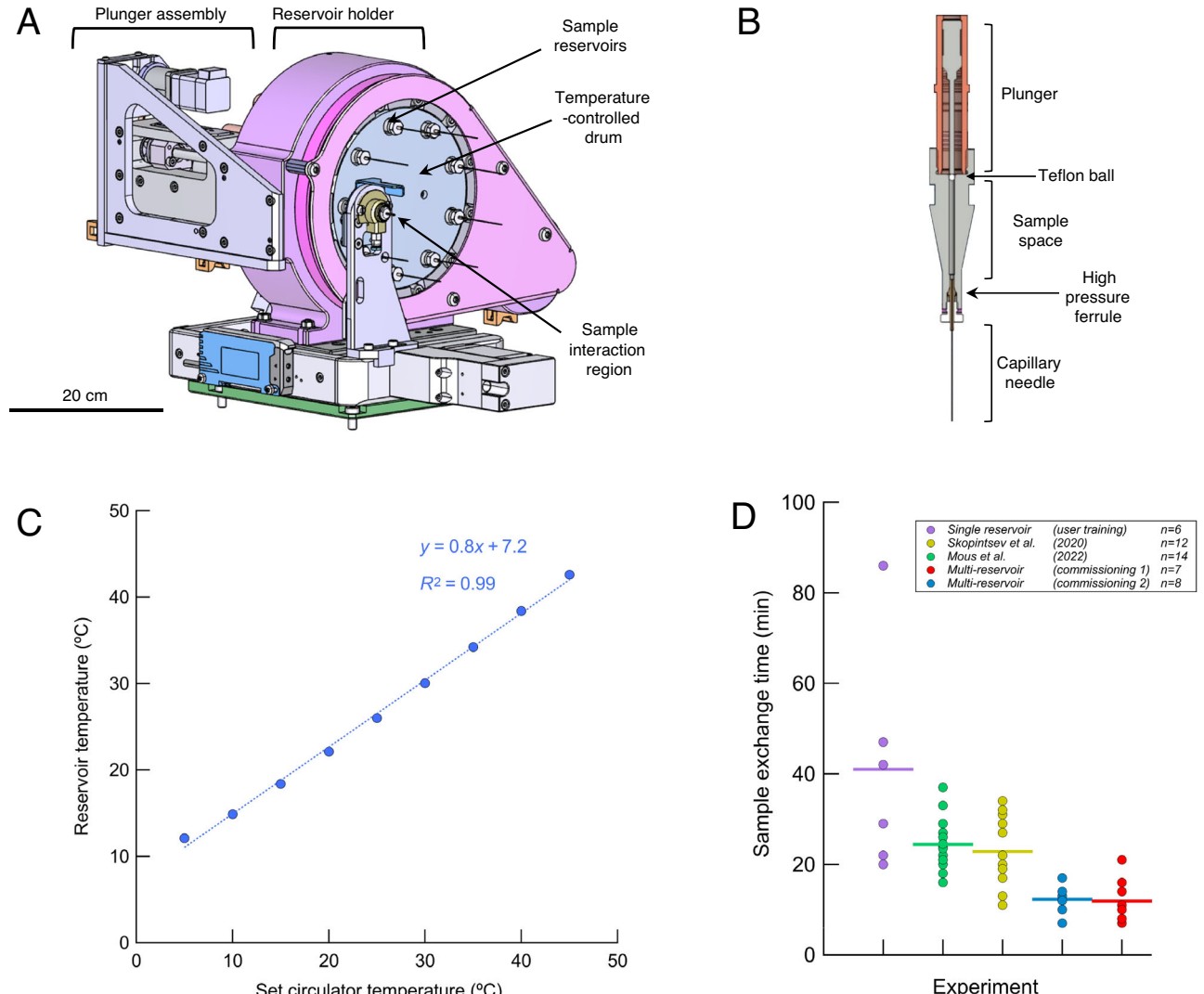

**Fig. 1 | Overview of the multi-reservoir high viscosity extruder. A** A schematic design of the device; the main component is a temperature-controlled rotating drum containing slots for nine individual sample reservoirs. The plunger assembly is used to control and monitor the sample ejection process. The device is installed on 3-axis translation stage. **B** Single sample reservoir (sample volume) with supporting components. **C** Temperature calibration of the device: horizontal axis indicates the set or demand temperature of the circulator while vertical axis displays the reservoir temperature measured in the sample space, indicated in panel **B**, dotted line is a linear data fit. **D** Sample exchange time comparison: each point indicates the time taken between the end and the start of data collection from consecutively loaded extruders *(n=number of loader extruders)*. The time required to setup the experimental hutch and condition the X-ray beam was excluded. Green *(n = 14)* and yellow *(n = 12)* points were determined using data published in Mous et al. 2022 and Skopintsev et al. 2020, respectively. Blue *(n = 7)* and red points *(n = 8)* present the exchange times of the two commissioning beamtimes described in this work. Purple points *(n = 6)* represent user training, courtesy of the SwissFEL Alvra experimental station staff.

experiment, the plunger was successfully driven at speeds of 1-30 μm/s for sample injection which corresponded to measured jet speeds of 0.9-30.9 mm/s when using a 75 μm capillary needle (with the limit on jet speed dictated by the pressure limit of the sample reservoir seals). See Supplementary Table 1 for a full data set on jet speeds. A video of high-speed recording of a sample extrusion at 8.4 mm/s, a speed typical in such experiments, is attached as Supplementary Movie 2.

Each sample reservoir assembly was built based on existing designs[15,23,33] (Fig. 1B). The loaded sample reservoir assemblies were fed manually into the individual slots of the reservoir holder where they lock into place via a spring-loaded bayonet mechanism. The loading sequence of the multi-reservoir (sample selection and movement of the chosen sample into the sample interaction region) takes approximately 60-120 seconds depending on the volume of sample loaded. This dependence on sample volume is simply a result of the motor having to translate over a larger distance in order to reach the sample if a smaller

volume is loaded. Including alignment of the X-ray beam to the extruded medium, it takes 11 minutes on average between stopping data collection on one reservoir and starting data collection on the next reservoir. Compared with our previous experience using a standard HVE injector[36,37] at the same experimental station, this accounts for a factor of two improvement in terms of sample exchange efficiency (Fig. 1D). We note that these two standard HVE injector experiments[36,37] can be considered as a benchmark executed by an experienced team involving at least three members. If we compare the multi-reservoir sample exchange times with the case of new users or user training, there is a factor of four improvement (Fig. 1D). Furthermore, once loaded with samples, the multi-reservoir can be controlled by a single operator via graphical user interface (Supplementary Fig. 5).

## Validation in experimental practice
In two commissioning beamtimes at the SwissFEL Alvra endstation, we validated the general performance of the multi-reservoir extruder and

**Table 1 | Summary of structures solved using the multi-reservoir high viscosity extruder grouped into three sections according to type of application: static, time-resolved and crystallographic drug screening**

| Crystal System | Abbreviated names of solved structures | Res† (Å) | $CC_{1/2}{}^a$ | $CC^{*a}$ | Multiplicity† | Ligand | Objectives and comments |
|---|---|---|---|---|---|---|---|
| Hen egg white lysozyme | Lyso (LCP) | 1.31 | 0.20 | 0.58 | 12.6 | — | Performance test using lipidic cubic phase carrier medium (Fig. 2A). |
| | Lyso (HEC) | 1.32 | 0.20 | 0.57 | 24.2 | — | Performance test using hydro-xyethylcellulose carrier medium (Fig. 2B). |
| Krokinobacter eikastus rhodopsin 2 | KR2 (dark) | 1.98 | 0.69 | 0.90 | 479.0 | — | Performance test of time-resolved study on KR2 (Fig. 2C). |
| | KR2 (light) time delay of 1µs | 2.04 | 0.63 | 0.88 | 224.5 | — | |
| tubulin-DARPin D1 complex: 1 copy of mammalian αβ-tubulin, 1 copy of the designed ankyrin repeat protein (DARPin) | TD1 (apo) | 2.10 | 0.61 | 0.87 | 43.8 | — | Detection of binding of a photochemical affinity switch designed for optical control of tubulin (Fig. 3A). |
| | TD1 (SBTubA4) | 1.80 | 0.76 | 0.93 | 133.8 | SBTubA4 | |
| $T_2R$-TTL complex: 2 copies of mammalian αβ-tubulin, 1 copy of the rat stathmin-like protein RB3, 1 copy of the chicken tubulin tyrosine ligase (TTL) | $T_2R$-TTL (apo) | 2.40 | 0.19 | 0.46 | 98.7 | — | Detection of low occupancy binding of photochemical affinity switches with applications in photopharmacology (Fig. 4). |
| | $T_2R$-TTL (SolQ2Br) | 2.60 | 0.26 | 0.64 | 251.9 | SolQ2Br | |
| | $T_2R$-TTL (Colchicine) | 2.70 | 0.21 | 0.59 | 90.8 | Colchicine | Screening of binding sites for approved drugs. |
| | $T_2R$-TTL (Epothilone A) | 2.60 | 0.37 | 0.73 | 137.4 | Epothilone A | |
| | $T_2R$-TTL (Vinblastine) | 3.10 | 0.26 | 0.64 | 31.9 | Vinblastine | |
| | $T_2R$-TTL (Ansamitocin P3) | 3.10 | 0.42 | 0.77 | 28.7 | Ansamitocin P3 | |
| | $T_2R$-TTL (Cocktail) | 2.40 | 0.32 | 0.69 | 177.6 | Mixture of: Epothilone A, Peloruside, Ansamitocin P3, Vinblastine and Colchicine | Simultaneous detection of ligand in multiple binding pockets (Fig. 3B). |

$^a$Note that the values are shown for the highest resolution shell, for more see Crystallographic Table 1 and 2.

determined 13 structures overall. These are summarised in Table 1 and in Crystallographic Table 1-2.

**Static structures.** Ever since the first structure of hen egg white lysozyme was determined in the 1960s[38], this protein has been used as a model system to develop new crystallographic methodology including the first high-resolution structures[39] and phasing experiments[40] at XFELs. Here we followed this tradition to demonstrate how the multireservoir extruder can be used in conjunction with viscous sample delivery media to resolve soluble protein structures at near-atomic resolution. Specifically, microcrystals of lysozyme were embedded into two viscous delivery media: lipidic cubic phase of monoolein (LCP) representative of a typical solution to grow and deliver membrane protein crystals[41] and hydroxyethylcellulose (HEC) as another established matrix for soluble proteins using a high-viscosity extruder[42,43]. Data collection was straightforward in both cases and permitted the determination of two structures at 1.3 Å resolution (Fig. 2A, B) limited by geometrical constraints in the chamber. Offering temperature control (Fig. 1C) and precise jet speeds, our device can be used with a large variety of delivery agents that have been tested in recent years[44,45] to optimise sample delivery. In particular, LCP exhibits a rich phase diagram[46,47] and its physical properties such as viscosity can be controlled using temperature as a tuning parameter.

**Time-resolved structures.** The light-driven sodium pump *Krokinobacter eikastus* rhodopsin 2 (KR2) was the target of one of our previous XFEL experiments[36] and was thus chosen to provide a reference for time-resolved data collection, examining structural changes after photoactivation of its retinal chromophore. We collected a complete time-resolved data set (light and dark) of 75,000 indexable images (Crystallographic Table 1) without having to enter the hutch or break the low-pressure helium environment of the Prime chamber. Xtrapol8[48] was used to calculate q-weighted difference maps $F_o{}^{light}$–$F_o{}^{dark}$ and corresponding extrapolated maps from these data (Supplementary Fig. 6). These maps depicted the structural changes 1 µs after retinal photoactivation (Fig. 2C). A comparison of the refined structure with the corresponding dark and light models[36] shows nearly identical movements of the retinal and the surrounding binding pocket. This result demonstrates that the multi-reservoir extruder can be used in time-resolved crystallography experiments where precise spatial and temporal calibration of the jet, optical laser beam, and X-ray beam are crucial for successful data collection.

**Crystallographic drug screening.** Molecular structures of proteins in complex with small molecule ligands provide valuable templates for SBDD. In order to explore the utility of the multi-reservoir extruder for efficient small molecule drug screening at XFELs, we have used two different crystals systems of the αβ-tubulin heterodimer: TD1[49] and $T_2R$-TTL[50,51]. Due to its key role during cell division, tubulin is a classical target for anti-cancer therapeutics including drugs such as Taxol and Vince alkaloids[52,53]. In addition, tubulin-binding molecules are well-known for their anti-inflammatory effect in gouty arthritis[54] and have recently been shown to decrease mortality in intensive care COVID-19 patients[55]. Notably, 27 ligand-binding sites have been identified in tubulin so far[56] making the protein an excellent target for SBDD.

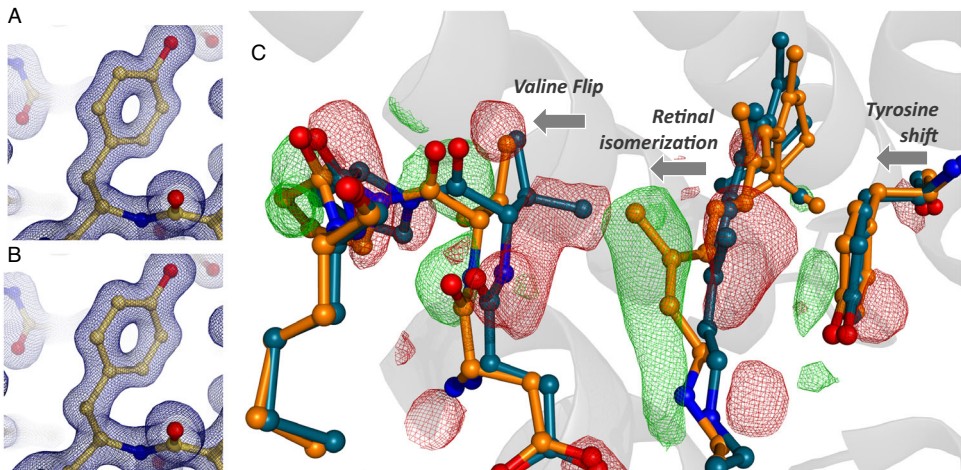

**Fig. 2 | Static and time-resolved structure determination.** Left panel: Static structure determination. Electron density ($2F_o$-$F_c$) around residue TYR53 of lysozyme using serial data collected with (**A**) LCP and (**B**) HEC as delivery. Blue grid representation indicates a sigma level of 1.5. Right panel: time-resolved photoactivation of KR2. **C** Difference electron density maps ($F_o^{Light}$–$F_o^{Dark}$) shown at sigma level of 3.0 depicting the structural changes of and around the retinal 1 µs after photoactivation.

As described above, the multi-reservoir extruder facilitates time-efficient data collection. To demonstrate how this could be exploited for SBDD, tubulin crystals incubated with single ligands (Table 1) were loaded into individual reservoirs. These reservoirs (loaded simultaneously into the multi-reservoir extruder) were sequentially measured without disrupting the low-pressure helium environment of the Prime chamber and without entering the beamline experimental hutch. For this part, we determined 7 crystal structures: apo structures for both TD1 and $T_2R$-TTL, $T_2R$-TTL complexed with four ligands known to bind to different tubulin sides, and TD1 complexed with the photochemical affinity switch SBTubA4[57]. SBTubA4 is a styrylbenzothiazole-based colchicine site binder that enables time-controlled, cell-precise optical control of microtubule dynamics and function not only in cell culture settings, but also over a range of multi-organ animal models[57]. The compound has numerous applications in cell biology as well as photopharmacology, and is a good candidate to study tubulin dynamics in TR-SFX experiments. We present the high-resolution structure of tubulin bound to SBTubA4 in Fig. 3A.

Another strategy to further increase the throughput of SBDD experiments is to use cocktails of molecules for soaking, instead of using solutions containing only a single compound[58]. We tested such an approach using the multi-reservoir extruder by collecting data from tubulin crystals incubated with a cocktail of five drugs, each targeting different binding sites, with known binding behaviour (Table 1). All five drugs included in the mixture could be resolved in their respective binding sites (Fig. 3B). This suggests that in the future, a combined approach relying on ligand cocktails and the multi-reservoir extruder could be used to efficiently and rapidly screen ligands in parallel and in a single XFEL experiment. Additionally, such cocktail approaches facilitate the exploration of allosteric and global effects whereby simultaneous ligand mixing might influence affinity levels at different binding sites as compared to mixing with single compounds (Supplementary Fig. 7).

**Detection and structural modelling of low occupancy binders**

Crystallographic fragment screening is a powerful method for identifying potential starting points for SBDD. Here, the affinity of the screened fragments for their protein targets is usually in the µM – mM range[59,60]. Given this, fragments of low-affinity can be challenging to be identified using X-ray crystallographic methods as low affinities are often associated with low occupancy ligand-bound states. Methods to identify low occupancy ligands have been successfully developed

for cryo-crystallography fragment screening approaches, namely Pandata-set density analysis (PANDDA)[61]. The parallels between identifying partial occupancy states in TR-SFX and low occupancy ligands using the PANDDA method has previously been noted[61,62]. Indeed, in TR-SFX isomorphous difference density maps and extrapolation of structure factors have permitted the modelling of conformational states that occur with occupancies of down to 10% within a crystal[48]. Here, we exploited serial crystallography data and associated data processing methods to identify a low occupancy ligand at room temperature.

We acquired data of tubulin soaked with a structurally unexploded photochemical ligand SolQ2Br, which is expected to bind the colchicine site. Here, we used the $T_2R$-TTL crystal system where there are two possible colchicine binding pockets available since the crystal system is formed of two copies of tubulin. Notably, the two sites can exhibit different ligand-occupancy levels and we thus refer to these as sites 1 and 2.

First, following practices in cryo-crystallography, we calculated the standard omit ($F_o$-$F_c$) maps based on the apo structure and refined electron density ($2F_o$-$F_c$) maps. We noted that for site 1, the standard omit map does not account for the apo structure revealing ligand binding (Fig. 4A) and the refined electron density revealed the ligand binding pose (Fig. 4B). However, for site 2 the signal level of the positive difference density was not significant enough to confirm ligand binding (Fig. 4C). Refining the observed data against a model containing two alternative conformations of ligand-bound and unbound states revealed predominantly the apo state (Fig. 4D).

Second, making use of techniques commonly used in TR-SFX we calculated isomorphous difference density ($F_o$-$F_o$) maps using the same datasets described above. The increased sensitivity of these difference maps allowed observing ligand binding not only in site 1, but also in site 2 (Fig. 4E). Furthermore, we extrapolated the structure factors of the ligand bound state to full occupancy using the software Xtrapol8[48]. Consequently, we could model the binding pose (Fig. 4F) and infer the occupancy level for site 2 as 25–30%.

With the successful application of serial crystallography approaches to data analysis, we wondered how many diffraction patterns need to be collected before the binding mode of the ligand can be unambiguously interpreted. To answer this question, we conducted a 'data titration' analysis by randomly selecting smaller and smaller subsets of indexed patterns from the ligand-soaked datasets.

In site 1, positive difference density, sufficient for observing the bound ligand, was present at a contour level of 3.0 sigma when

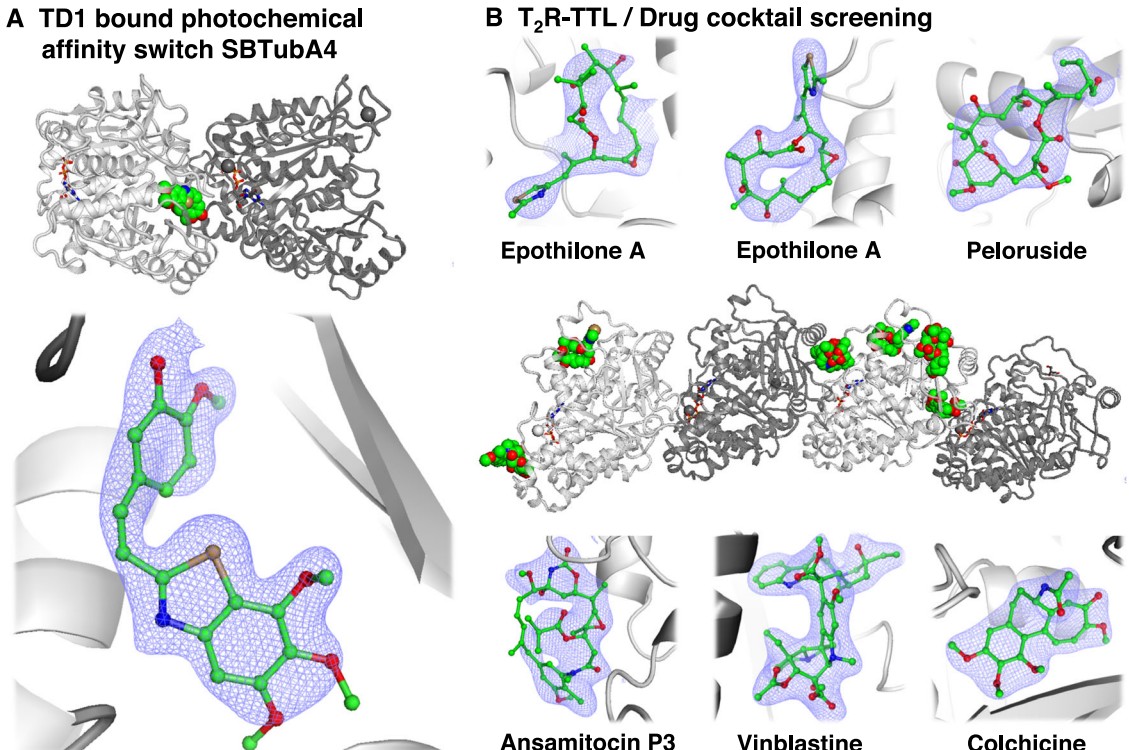

**A** **TD1 bound photochemical affinity switch SBTubA4**

**B** **T$_2$R-TTL / Drug cocktail screening**

Epothilone A    Epothilone A    Peloruside

Ansamitocin P3    Vinblastine    Colchicine

**Fig. 3 | Tubulin ligand screening. A** Binding pocket and high-resolution crystal structure of the photochemical affinity switch SBTubA4 for the optical control of tubulin dynamics. Presented electron density (2F$_o$-F$_c$) map is shown at a sigma level of 1.0. **B** Tubulin/Drug cocktail screening: Binding pockets and crystal structures of known anti-tubulin drugs Epothilone A, Peloruside, Ansamitocin P3, Vinblastine, and Colchicine. Presented electron density (2F$_o$-F$_c$) maps are shown at a sigma level of 1.0.

comparing 1000 randomly selected diffraction patterns against an apo tubulin data set containing 10,000 indexed patterns. This was the case for both standard omit (F$_o$-F$_c$) maps (Fig. 5A) and isomorphous difference density (F$_o$-F$_o$) maps (Fig. 5B). The map quality was sufficient to place the ligand in the binding site and was in line with our previous serial synchrotron experiments on tubulin[22]. Full 'data titration' analysis, including both (F$_o$-F$_c$) and (F$_o$-F$_o$) maps resulting from subsets of 5000, 4000, 3000, 2000, 1000, and 500 indexed patterns, is presented in Supplementary Fig. 8. Comparable 'data titration' analysis for standard omit (F$_o$-F$_c$) maps has also been shown for several small molecules by Moreno-Chicano et al.[63]

For site 2, unlike standard omit (F$_o$-F$_c$) maps (Fig. 5C), only isomorphous difference density (F$_o$-F$_o$) maps revealed the ligand even when as few as 1000 images were considered (Fig. 5D). Obtained standard omit (F$_o$-F$_c$) maps did not provide a similar quality regardless of the number of considered patterns (Supplementary Fig. 9). Reliable detection of fragment binding for both high and low occupancy binders is therefore clearly improved by calculating isomorphous difference density (F$_o$-F$_o$) maps from serial crystallography data and, in this way, even relatively few images are sufficient to detect binding. Assuming a conservative overall indexing rate of 10%, such data can be collected in approximately 100 seconds. Crystallographic fragment-based drug discovery at synchrotrons has been very successful for tubulin[56]; however, until now, only strong binders with an occupancy above 50% have been detected using conventional methods[56]. The results here suggest that using serial crystallography data can push this limit substantially lower, with the added advantage of these experiments being conducted at room temperature.

## Discussion
This work demonstrates the ability of the multi-reservoir high viscosity extruder to efficiently perform structural biology experiments

at XFELs. Specifically, we were able to: 1) determine high-quality structures of soluble and membrane proteins using different viscous carriers; (2) perform time-resolved pump-probe studies; and (3) successfully screen for low occupancy binders of tubulin. Since high-viscosity extruders are the most common delivery method for serial crystallography at XFEL sources[2], we expect our device to drive and improve the scientific outcome of serial crystallography experiments. The simplified handling has the potential to lower the barrier for conducting XFEL measurements (particularly for inexperienced users), since a) constant simultaneous sample preparation during the beamtime experiment can be avoided; b) the automated sample loading procedure reduces the risk of interfering with the laser and X-ray beam alignment; and c) data can be collected more efficiently. Additionally, temperature control of the sample offered by the device permits fine-tuning of jetting behaviour (e.g., through optimizing phase composition and viscosity) and would permit the collection of temperature-dependent intermediate conformational states.

To this day TR-SFX remains the most developed and successful technique to visualise protein dynamics down to ultrafast timescales. These timescales still remain inaccessible using other methods, as discussed elsewhere[2–4,22,64–66], whether experimental, such as time-resolved cryo-EM (limited to ms[67–69]), or computational such as AlphaFold[70]. Further efficiency gains during sample injection and automation of sample delivery will make the technique more accessible to a wider user community and will drive progress towards answering a wider range of biochemical questions. Development and demonstration of the multireservoir extruder is an initial step in this direction.

Macromolecular X-ray crystallography has played a key role in SBDD, with particular gains being made from high-throughput screening pipelines such as those offered by XChem at Diamond

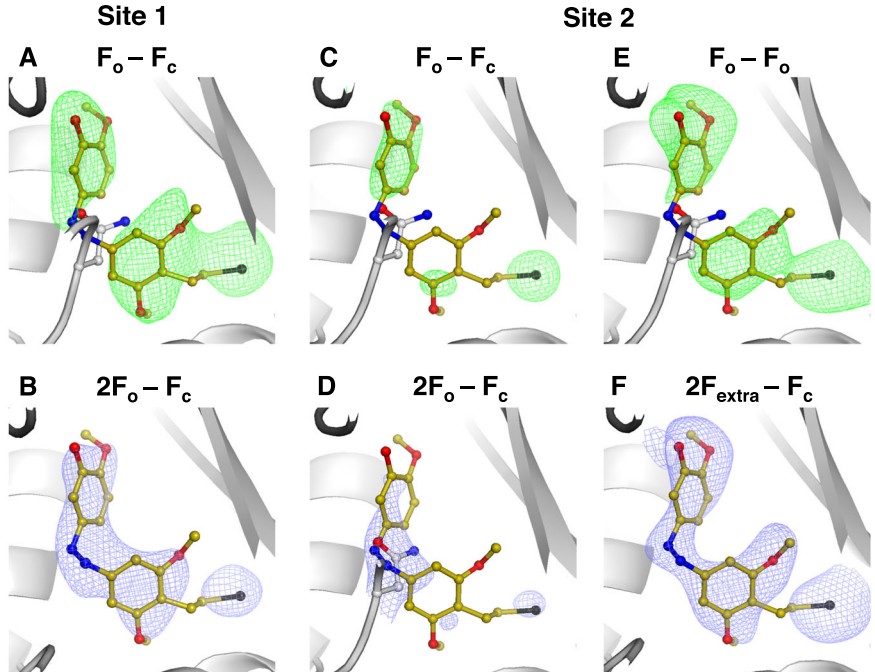

**Fig. 4 | Comparison of different data analysis approaches for detecting low occupancy binding of SolQ2Br.** The chosen $T_2R$-TTL system comprises two binding sites for the photochemical affinity switch SolQ2Br and we refer to those as: high occupancy site 1 (left panel: A and B) and low occupancy site 2 (right panel: **C, D, E** and **F**). **A** Standard omit ($F_o$-$F_c$) map against apo protein state. **B** Refined electron density ($2F_o$-$F_c$) map. **C** Standard omit ($F_o$-$F_c$) map against apo protein state. **D** Electron density ($2F_o$-$F_c$) map refined against a model containing alternative conformations of ligand bound and apo protein state. **E** Isomorphous difference density ($F_o$-$F_o$) map against apo protein state. **F** Extrapolated electron density ($2F_{extra}$-$F_c$) map; the ratio of ligand bound state was determined to ~30%. The SolQ2Br structure is overlayed in all shown standard omit maps and isomorphous difference density maps for better visualization. The apo protein state is presented by the flipped in residue Asn247 of site 1 or 2 respectively. For comparability, all standard omit maps and isomorphous difference density maps are shown at a sigma level of 3.0, all electron density maps are shown at a sigma level of 1.5.

Light Source[71], FragMAX at MAX-IV[72,73], the EMBL HTX Facility in Grenoble[74], and FFCS at the Swiss Light Source[75]. Even with the rapid advances in structure prediction[76] and cryo-electron microscopy[77], SBDD will remain an important application of X-ray crystallography for the foreseeable future. Due to restricted availability and operating costs, attempting SBDD using structural data from XFELs has not spread beyond specific cases such as difficult-to-crystallize G protein-coupled receptors[77]. Nevertheless, SBDD using serial crystallography at the 4th generation synchrotrons and XFELs can develop into an important application that will benefit from developments in sample delivery methods[19,78–81].

Efficient collection of serial crystallography data makes it reasonable to screen fragment and compound binding at XFELs. Notably, in crystallographic fragment screening approaches, the majority of fragments bind with less than 70% occupancy[61]. As suggested in the literature and in combination with our device, we showed here that serial crystallography approaches are able to detect low occupancy binders (25–30%) at room-temperature. Just 1000 images were sufficient to detect reliably binders of low occupancy via isomorphous difference density maps and extrapolation of the bound protein states allowed accurate modelling of the respective binding poses. Furthermore, conducting crystallographic fragment screening experiments at room-temperature has been shown to influence ligand binding modes[82]. Therefore, serial approaches will complement existing cryo-crystallography pipelines whilst offering unique information as well. We anticipate that further improvements in the efficiency of serial crystallography at XFELs will allow the method to grow into a promising tool for SBDD including studies of the dynamics of time- and temperature-resolved ligand binding and release.

## Methods

### SwissFEL Sample Preparation

**Lysozyme.** Lyophilized lysozyme was purchased from Sigma-Aldrich (specification L2879) and crystallised using a batch-crystallization approach. The lyophilized lysozyme was first resolved in 100 mM sodium acetate pH 3.0 to a final concentration of 25 mg/mL. In a 1:1 ratio, the prepared solution was then added to a precipitation solution containing 28% (w/v) sodium chloride, 8% (w/v) PEG 6000, and 100 mM sodium acetate pH 3.0. Crystals grew overnight at 20 °C and presented size dimensions of approximately 2 × 2 × 5 µm. For LCP and HEC incorporation, a 100 µL aliquot of lysozyme crystal slurry was pelleted at 500x g for 60 s and the supernatant removed. Incorporation into the LCP was achieved by first preparing 'empty' LCP using monoolein in a 2:3 v/v ratio with 100 mM sodium acetate pH 3.0. The crystal pellet was then added to the 'empty' LCP followed by a volume of monoolein corresponding to 1.5 times the volume of added crystal slurry. Additional small aliquots of monoolein were added to bring the preparation into a clear LCP. Incorporation into HEC was carried out using a 3-way syringe coupler[83]. The crystal slurry was mixed in a 1:6 volumetric ratio with 22% (w/v) HEC prepared in water.

**Krokinobacter eikastus rhodopsin 2 (KR2).** Protein expression and purification was based on previously described experiments[36]. Protein crystallization of KR2 samples took place in LCP under conditions similar to those described in literature[36]. LCP was formed by mixing protein buffer and monoolein in a 4:7 v/v ratio through coupled gas-tight Hamilton syringes. The formed LCP was extruded through Hamilton needles into plastic syringes loaded with precipitant solution containing 34% (w/v) PEG 200, 200 mM sodium acetate pH 4.4, and 150 mM MgCl$_2$. Protein crystallization occurred overnight in the dark

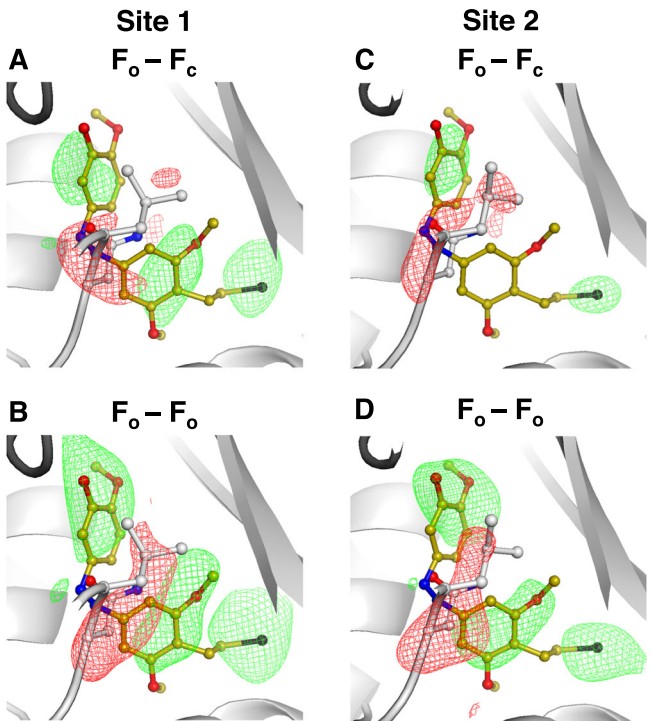

**Fig. 5 | Comparison of different data analysis approaches for detecting low occupancy binding.** The chosen $T_2$R-TTL system comprises two binding sites for the photochemical affinity switch SolQ2Br and we refer to those as: high occupancy site 1 (left panel: **A** and **B**) and low occupancy site 2 (right panel: **C** and **D**). Standard omit ($F_o$-$F_c$) maps against apo protein state are shown in (**A**) and (**C**); isomorphous difference density ($F_o$-$F_o$) maps against apo protein state are shown in (**B**) and (**D**). All maps are calculated based on a data set containing 1000 randomly selected indexable diffraction patterns. The SolQ2Br structure is overlayed in all shown standard omit maps and isomorphous difference density maps for better visualization. The apo protein state is presented by the flipped in residue Asn247 of site 1 or 2 respectively. For comparability, all standard omit maps and isomorphous difference density maps are shown at a sigma level of 3.0. Positive difference density is shown in green; negative difference density is shown in red.

at 20 °C and yielded plate-like blue KR2 crystals with size dimensions of 20–35 × 20–35 × 1–3 μm.

For SwissFEL sample preparation, the precipitant solution was washed out by soaking the LCP twice in a solution containing 34% (w/v) PEG 200 and 150 mM NaCl. The washed phase was harvested into Hamilton syringes in 60 μl fractions, and doped with 33 μl monoolein and 3.0 μl 50% (w/v) PEG 1500 to form a stable jetting phase. Before data collection, the crystal phase was mixed with LCP prepared from monoolein and a solution of 34% (w/v) PEG 200, 1 M Tris pH 9.0, and 150 mM NaCl through a 3-way syringe coupler[83]. The volumes of the mixed phases were picked such that the water fraction of the meso-phase would contain 34% (w/v) PEG 200 and PEG 1500, 200 mM Tris, and 150 mM NaCl. The applied procedure changed the former blue-colored KR2 crystals into red-colored crystals.

**Tubulin.** Tubulin DARPin D1 (TD1) protein complex: first described by Pecqueur et al.[49], detailed descriptions and protocols of how to express, purify and prepare necessary proteins and how to formulate the TD1 tubulin complex is presented by Mühlethaler et al.[84]. The large-scale protein production essential for serial crystallography experiments at XFELs is described as well.

Concentrated crystals were incubated with the non-binding *trans*-SBTubA4 at a concentration of 2.5 mM and then illuminated for 10 minutes at 385 nm to induce majority ligand isomerization towards the binding *cis* isomer. For both measured TD1 samples (the SBTubA4

incubated, and the compound-free apo protein state), incorporation into HEC was carried out using a 3-way coupler[83]. In the process, crystals were crushed to a final size of 20 × 5 × 5 μm. The respective crystal slurry was gently mixed in a 1:1 volumetric ratio with 22% (w/v) HEC prepared in water.

Tubulin $T_2$R-TTL protein complex: detailed descriptions and protocols of how to express, purify and prepare necessary proteins and how to formulate the $T_2$R-TTL tubulin complex is presented by Mühlethaler et al.[84]. For the large-scale protein crystal production essential for serial crystallography experiments at XFELs, the $T_2$R-TTL tubulin complex was concentrated to 20 mg/mL. In PCR tubes, first 4 μL of protein complex solution was provided and then carefully mixed with additional 6 μL of the precipitant solution containing 6% (w/v) PEG 4000, 30 mM $MgCl_2$, 30 mM $CaCl_2$, 100 mM MES/Imidazole, and 5 mM L-tyrosine. The PCR tubes were sealed with Parafilm and stored for two days at room temperature. The formed pellet of crystals with a homogenous size distribution of 30 × 5 × 5 μm was carefully resuspended and several setups were combined in a 1.5 mL Eppendorf tube. After the crystals have settled again, the supernatant was removed and the crystals used for compound incubation without any further concentration.

For the single compound structures, concentrated crystals were incubated for 45 minutes with 1 mM of respective commercially available approved anti-tubulin drugs, namely colchicine, epothilone A, vinblastine, and ansamitocin P3. For the photochemical affinity switch SolQ2Br, 4 mM of inactive *trans* isomer were added and the concentrated crystals then illuminated for 45 minutes at 385 nm to drive majority isomerisation to the active *cis* isomer. For the cocktail structure, 1 mM each of colchicine, epothilone A, vinblastine, peloruside A, and ansamitocin P3 were added to the concentrated crystal solution. The crystals were then immediately prepared for injection into the X-ray beam. For all described $T_2$R-TTL crystal systems, 20 μL of concentrated crystals were homogenously and gently embedded into 20 μL of a 22% HEC matrix using Hamilton syringes and a three-way-coupler.[83]

## SwissFEL data collection

**Serial protein crystallography.** These data were collected in February 2020 at the SwissFEL Alvra experimental station. To reduce X-ray background scattering, the sample chamber was equilibrated to a 200-500 mbar helium environment. The data from lysozyme and tubulin crystals were collected at a photon energy of 12.0 keV with a pulse energy of 170 μJ. The pulse length was approximately 70 fs and was delivered at a repetition rate of 50 Hz. The X-ray intensity was adjusted using solid attenuators to maximize the diffraction signal without disrupting the sample injector flow or damaging the detector. The X-ray beam was focused using Kirkpatrick-Baez (KB) mirrors to a spot size of approximately 5 μm by 5 μm (vertical by horizontal, FWHM). Data were recorded using a Jungfrau 16 M detector in 4 M mode.

**Time-resolved serial femtosecond crystallography.** These data were collected in September 2021 at the SwissFEL Alvra experimental station. To reduce X-ray background scattering, the sample chamber was equilibrated to a 200-500 mbar helium environment. The data from KR2 crystals were collected using a photon energy of 12.0 keV with a pulse energy of 440 μJ. The pulse length was approximately 70 fs and was delivered at a repetition rate of 100 Hz. The X-ray beam was focused using KB mirrors to a spot size of approximately 2 μm by 2 μm (vertical by horizontal, FWHM). Data were recorded using a Jungfrau 16 M detector in 4 M mode. The probing XFEL beam intersected with a circularly polarized pump laser. The laser pulses were of 150 fs duration, 575 nm wavelength and 5.7 μJ total energy in a focal spot of 90 × 90 $μm^2$ beam ($1/e^2$).

## Crystallographic data processing

All recorded serial crystallography data were indexed, integrated and merged using different versions of Crystfel[85,86]. Structural refinements

of the models were done using PHENIX[87] with iterative cycles of manual adjustments made in Coot[88]. All figures were created using the PyMOL Molecular Graphics System, Version 2.0 Schrödinger, LLC.

**Lysozyme.** All data were indexed, integrated and merged using Crystfel version 0.9.1.[85,86] Data were indexed using xgandalf[89]. Data were integrated using the --rings option in indexamajig. Patterns were merged using partialator with the following options: --model = unity, --iterations = 3. Patterns were collected for lysozyme samples in LCP and HEC with corresponding hit rates of 23% and 43% respectively. To compare data quality, 25000 indexable patterns from both datasets were used and merged respectively to produce the crystallographic refinement table. The staraniso server was used for anisotropic correction of obtained data[90]. The difference in the hit rates originates from the nature of the sample carrier (HEC allows for much higher densities of crystals as compared to LCP).

**Krokinobacter eikastus rhodopsin 2 (KR2).** In the experiment, full dark data was collected first and then light data was obtained (registered ratio 1:1 – light:dark). Indexing, integrating and merging of obtained tubulin crystal data was performed using Crystfel version 0.9.1.[85] In detail, the peakfinder8 algorithm was used for peak detection, and xgandalf[89] was used to index obtained data. Peaks were integrated using --threshold=450, --int-radius=3,4,8 --SNR = 4.0 --min-pix-count=2 --min-peaks=8 --tolerance=5,5,5,2.5. The partialator options --model=unity, --iterations=1.0, --push-res=1.5 were used to merge selected patterns. The staraniso server was used for anisotropic correction of obtained data[90].

Difference density maps calculation: electron difference maps between light and dark KR2 datasets were calculated with Xtrapol8, using q-weighting, with data ranging from 1.7 Å to 7 Å. Extrapolation was also performed with Xtrapol8, using q-weighting and data ranging from 2.2 Å to 15 Å. The activation level was determined visually to be around 19%. A comparison of the refined structure with the corresponding dark and light models[36] was done using PDB codes: 6TK6 (Femtosecond to millisecond structural changes in a light-driven sodium pump: Dark) and 6TK2 (Femtosecond to millisecond structural changes in a light-driven sodium pump: 1 ms).

**Tubulin.** Indexing, integrating and merging of obtained tubulin crystal data was performed using Crystfel version 0.8.0.[85,86] In detail, the peakfinder8 algorithm was used for peak detection, xgandalf[89] was used to index obtained data.

For TD1, peaks were integrated using --threshold=350, --ring-radius=2,3,5 --SNR = 3.75 --min-pix-count=2 --min-peaks=10 --high-res=1.6; partialator options --model=unity, --iterations=1.0, --push-res=1.5 were used to merge selected patterns. The staraniso server was used for anisotropic correction of obtained data[90]. The structure of the apo protein state was solved by molecular replacement with 5NQT (Tubulin Darpin room-temperature structure) as a search model. The *cis*-SBTubA4 bound protein state was solved by molecular replacement with the apo structure as a search model. The ligand restraints were generated with the grade server[90] using their SMILES annotation.

For T$_2$R-TTL, peaks were integrated using --threshold=200, --ring-radius=2,3,5 --SNR = 4 --min-pix-count=2 --min-peaks=12 --highres=1.6; partialator options --model=unity, --iterations=1.0, --push-res=1.5 were used to merge selected patterns. The structure of the apo protein state was solved by molecular replacement with 4O2B (Tubulin-Colchicine complex) as a search model. The T$_2$R-TTL cocktail structure was solved by molecular replacement with the apo structure as a search model. The T$_2$R-TTL colchicine structure was solved by molecular replacement with 4O2B (Tubulin-Colchicine complex) as a search model. The T$_2$R-TTL vinblastine structure was solved by molecular replacement with 5J2T (Tubulin-Vinblastine complex) as a search model. The

T$_2$R-TTL ansamitocin P3 structure was solved by molecular replacement with 7E4P (Crystal structure of tubulin in complex with Ansamitocin P3) as a search model. The T$_2$R-TTL epothilone A structure was solved by molecular replacement with 4O4I (Tubulin-Laulimalide-Epothilone A complex) as a search model. All known commercially available approved anti-tubulin drugs structures and restraints were taken from PHENIX tool eLBOW[91]. The T$_2$R-TTL SolQ2Br structure was solved by molecular replacement with the apo structure as a search model. The ligand and restraints were generated with the grade server[92] using their SMILES annotation.

Difference density maps calculation: calculations of F$_o$$^{light}$ – F$_o$$^{dark}$ difference maps were performed using PHENIX v1.19.255 based on our apo protein data set containing 10,000 indexable images. Briefly, the multi-scaling option was used excluding amplitudes smaller than 3 sigma. For all calculated F$_o$$^{light}$ – F$_o$$^{dark}$ difference maps, the phases of the respective refined apo states were used.

## Reporting summary
Further information on research design is available in the Nature Portfolio Reporting Summary linked to this article.

## Data availability
Different PDB entries were used to perform molecular replacement (see Data Processing). Coordinates and structure factors of the presented serial crystallographic SwissFEL structures have been deposited in the PDB database under accession codes: 8CL5 (Lysozyme, embedded in LCP), 8CL6 (Lysozyme, embedded in HEC), 8CL7 (KR2, dark state), 8CL9 (Tubulin (TD1), apo state), 8CLC (Tubulin (T2R-TTL), apo state), 8CLF (Tubulin (T2R-TTL), SolQ2Br bound state), 8CLB (Tubulin (T2R-TTL), Colchicine bound state), 8CLG (Tubulin (T2R-TTL), Epothilone A bound state), 8CLE (Tubulin (T2R-TTL), Vinblastine bound state), 8CLD (Tubulin (T2R-TTL), Ansamitocin P3 bound state), 8CLH (Tubulin (T2R-TTL), drug cocktail). The model refined against extrapolated data, extrapolated structure factors, and light data used for data extrapolation have been deposited under accession codes 8CL8 (KR2, 1μs time delay). Other data are available from the corresponding authors upon request. The source data underlying Fig. 1c, d are provided as a Source Data file. Source data are provided with this paper.

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

## Acknowledgements

We would like to thank Gebhard Schertler and Rafael Abela for support and helpful discussions. Peloruside A and epothilone A were kind gifts from Peter Northcote and John Miller, and Karl-Heinz Altmann, respectively. The project was supported by grants from the Swiss National Science Foundation: 310030_192566, to M.O.S.; 31003A_179351 to J.S.; and 310030_207462, to J.S.. O.T.-S. acknowledges funding support from the German Research Foundation (DFG) through Emmy Noether grant TH2231/1-1 (project number 400324123). We would like to acknowledge Kurt Bitterli and his consultation of the technical aspect of design and user-device interaction as well as graphical user interface development. We thank the Macromolecular Crystallography group for support and we further thank everybody involved in ensuring the smooth operation of the Swiss X-ray free-electron laser. Finally, we thank the electrical and mechanical workshops at PSI who helped building the device.

## Author contributions

The project was initiated and coordinated by J.S. D.J., C.M., J.S., D.G. designed the prototype device and helped with subsequent interactions. M.S., D.G. and D.J. developed Computer Aided Models (CAD) models. D.J., E.V.B., M.W.K., M.W., A.F., C.Ba., C.C., P.J.M.J., and F.D. helped with offline testing of the device including overall performance preference and machine protocol improvements. M.W.K. and E.V.B performed temperature control calibration, characterized jetting conditions and obtained high-speed camera recordings of the extrusion process. S.S. and T.Z. developed graphical user interface for the device. K.K. and L.G. synthesized and characterised the photoactive affinity switches. Protein purification and crystallisation was conducted by A.F., Q.B., R.T.S. and M.W. M.W., Q.B. and T.W. analysed the crystallographic data. D.O and K.N. set up the pipeline for on-the-fly data processing and acquisition followed by manual optimization by Q.B., M.W. and T.W. H.G., M.C., Y.K., R.T.S., G.K., C.C., P.J.M.J., F.D., J.B., C.Ba., K.N. collected the X-ray diffraction data and helped with the preparation of the experimental hutch including laser characterisation and alignment. M.O.S. supervised the tubulin work. O.T.-S. lead and supervised chemical synthesis work on photoactive switches. C.Bo. supervised the Alvra scientists as well as beamtime planning. J.S. supervised SFX work. M.W., M.W.K., E.V.B., Q.B., M.O.S. and J.S. wrote the manuscript. All the authors read the manuscript and agreed to its contents.

## Competing interests

The authors declare no competing interests.

## Additional information

¹Laboratory of Biomolecular Research, Division of Biology and Chemistry, Paul Scherrer Institut, Villigen-PSI, Villigen 5232, Switzerland. ²Laboratory for Synchrotron Radiation and Femtochemistry, Photon Science Division, Paul Scherrer Institut, Villigen-PSI, 5232 Villigen, Switzerland. ³Laboratory of Nanoscale Biology, Photon Science Division, Paul Scherrer Institut, Villigen-PSI, 5232 Villigen, Switzerland. ⁴Laboratory for Macromolecules and Bioimaging, Photon Science Division, Paul Scherrer Institut, Villigen-PSI, 5232 Villigen, Switzerland. ⁵Scientific Computing, Theory and Data Division, Paul Scherrer Institut, Villigen-PSI, 5232 Villigen, Switzerland. ⁶Laboratory for Nonlinear Optics, Photon Science Division, Paul Scherrer Institut, Villigen-PSI, 5232 Villigen, Switzerland. ⁷Large Research Facilities Division, Paul Scherrer Institut, Villigen-PSI, 5232 Villigen, Switzerland. ⁸Department of Pharmacy, Ludwig-Maximilians University of Munich, Butenandtstr. 7, Munich 81377, Germany. ⁹LUXS Laboratory for Ultrafast X-ray Sciences, Institute of Chemical Sciences and Engineering, École Polytechnique Fédérale de Lausanne (EPFL), CH-1015 Lausanne, Switzerland. ¹⁰Biozentrum, University of Basel, 4056 Basel, Switzerland. ¹¹Femtosecond X-ray Experiments Instrument, European XFEL GmbH, Schenefeld, Germany. ¹²These authors contributed equally: Maximilian Wranik, Michal W. Kepa, Emma V. Beale. ✉e-mail: maximilian.wranik@psi.ch; mwranik@stanford.edu; michal.kepa@psi.ch

