## [Peer Review File · Nature Communications]

A multi-reservoir extruder for time-resolved serial protein crystallography and compound screening at X-ray free-electron lasersReviewers' Comments:

Reviewer #1:

Remarks to the Author:

The manuscript by Wranik, Kepa, Beale et al. is excellent and provides a significant step forward in using HVE injection for serial crystallography and thus opening new possibilities in time-resolved structural biology and structure-based drug discovery. It reads very well, is scientifically sound and I fully support publication in Nature communications after some corrections:

1) In the abstract the authors write "and offered a path forward for the observation of temperature-dependent structural changes." This sounds to me as if temperature-dependent changes were probed and have been observed in the work presented here, which is not the case. I would suggest to re-phrase this, as to not confuse readers of the article.

2) Line 27. The authors are talking about experiments using HVE injectors and are citing "Mehrabi, P. et al. Time-resolved crystallography reveals allosteric communication aligned with molecular breathing. *Science*. 365, 1167–1170 (2019).", which is an excellent example for TR-SSX, however no HVE injection was used to obtain the results presented.

3) Line 29ff only names 4th generation Synchrotrons/upgrades in Europe and not even all of them. NSLS-II is already in operation, the APS upgrade is on-going and in Europe the PETRAIV upgrade will happen only slightly after the Diamond-II upgrade.

4) Line 31/32 this is only true, if HVE injectors are widely used in SSX. I would re-phrase this a bit.

5) I would simply cut this sentence: "We note that in terms of sample delivery, the higher flux of the 4th generation synchrotrons will present challenges similar to those at XFELs, e.g., handling a large number of micron-size crystals."

6) Line 89 -91: "A guiding aperture then ensures an accurate fit of the capillary needle of the sample reservoir into the ceramic gas aperture. The gas aperture is connected to a helium gas supply fitted with a remotely controlled inline pressure regulator." How is the ceramic gas aperture sealed towards the back (the exchangeable sample capillary). Would be good to elaborate a bit on that.

7) Line 118-120: "The loading sequence of the multi-reservoir (sample selection and movement of the chosen sample into the sample interaction region) takes approximately 60-120 seconds depending on the volume of sample loaded." Why does this depend on the volume of sample loaded?

8) Line 157-158: "These data were used to determine difference maps F_{light}-F_{odark} and extrapolated maps (Supplementary Figure S6)," what kind of isomorphous difference maps are you using? q-weighted generated with Xtrapol8? Then you should mention this here and also cite Xtrapol8 already here. Same is true for all other F_o-F_o maps. Please always mention if they are standard F_o-F_o maps or q- or k-weighted (for example line 250 and so on).

9) Line 226: should be "difference density" not "difference denisty"

10) Line 436-442: the information about search models is important, but could be included in the crystallographic table and omitted here.

11) Figure 1: a schematic, like in panel B, but for the whole assembly, including the interaction region would be good to understand better the set-up.

12) Figure 2: what is the reason for adding a second 2F_o-F_c map at a sigma-level of 3.6? It would look better without it and gives not much additional information.

13)Figure 4: "and serial" instead of "andserial" (caption line 1)

14)Figure 5:" Comparison of conventional and serial protein crystallography". What does conventional mean here? Did you collect additional data using standard cryo-MX? If not, please remove "conventional" and re-phrase

15)Table 1: what about CC* in addition to CC1/2? Also, you mention use of Staraniso for KR2, what do the "aniso" data processing stats for lysozyme refer to? Also you mentioned you used Staraniso for TD1, however there is no "aniso" column for the TD1-datasets.

16)Supplementary figure S2: it would be good to label all the parts. Panels A and B could also be moved to main and merged with figure 1.

Reviewer #2:

The paper by Wranik, Kepa, Beale et al. reports on a new multi-reservoir extruder device for serial presentation of crystalline samples at X-ray Free Electrons lasers and synchrotrons. The idea of a rotating barrel to store sample cartridges is appealing, as is the possibility of collecting data at varying temperature, e.g. to obtain insights into equilibrium dynamics. The authors also did a good job at demonstrating that their system is suited for static, time-resolved and ligand based crystallographic studies, and this referee has no doubt that their system will be used and useful by/to the (hopefully-growing) serial-crystallography community. However, it is unclear what in this paper is novel enough to warrant publications in Nature Communications. Indeed, (i) the lysozyme structure is evidently not new; (ii) the KR2 results are not new ; (iii) the binding sites on tubulin were all already known; (iv) the 'evolution' of the extruder is incremental. Maybe would the feeling of not learning much have been less present if insights into equilibrium dynamics had been by obtained by collection of data at varying temperatures, or if the pump-probe experiment had been carried out on the complex of tubulin with the photochemical affinity switch SBTub4. But this is not the case, and therefore the existence of this device is the only thing one learns from the paper. The referee appreciates the hard work of the authors, congratulates them for the successful and prolific commissioning of their device, and wishes not to undervalue the potential of their multi-reservoir extruder device. However, it is the sincere belief of this referee that this paper would be more suited in a specialised journal such as IUCR J, J Synchr Rad or Acta Cryst D.

Before resubmission of the paper (to this journal or another), the authors should devote some time to proof-reading. Notably, they should verify their reference list:

- in some references, some authors are listed with their full name, others with their surname in full but initials for the first name, and yet others with only initials (see e.g. ref. 2)
- In other references, we have the name of the website that is printed along with that of the paper (ref 3)
- Other references have new words in their title (e.g. ref 6: « research papers » is appended to the title.
- Other references have authors listed in an incorrect order (e.g. ref 58)
- Some references are duplicated (e.g. ref 79 and 81).

The authors should correct for these mistakes (which are not only details) as they leave the reader with the bad impression of an hastily written paper...

Other places where attention could be given to wording;

« With the advent of diffraction limited 4th generation synchrotrons (and dedicated beamlines) including: MAX IV (BioMAX)²⁷ and ESRF₃₀ (ID29)²⁸ both in operation as well as Diamond-II and the SLS2.0, planned operation in 2027 and 2025, respectively, it is evident that serial crystallography methods, including HVE injectors, are becoming even more widespread. » ==> the authors should rephrase that sentence.

« We note that in terms of sample delivery, the higher flux of the 4th generation synchrotrons will present challenges similar to those at XFELs, e.g., handling a large number of micron-size crystals. » ==> the authors should explain why.

« In this work, we describe a new multi-reservoir device for viscous sample extrusion, which was designed to address some of the limitations of existing HVE injectors^{15,23,29} as well as to automate the sample exchange process in a serial crystallography experiment. » ==> The authors could list some of these limitations so that it is clear which of these are addressed by their new design (and which aren't).

« These reservoirs (loaded simultaneously into the multi-reservoir extruder) were sequentially measured without disrupting the evacuated Prime chamber or beamline hutch. » ==> the authors should rephrase that sentence, notably the term « evacuated Prime chamber ».

« This information cannot easily be obtained by any other means^{2-4,22,60-62} both experimentally, such as time-resolved cryo-EM where time resolution is limited to tens of ms^{63-65,275} and computational such as AlphaFold⁶⁶. » ==> the authors should rephrase that sentence.

« This suggests that in the future, a combined approach relying on ligand cocktails and the multi-reservoir extruder could be used to efficiently and rapidly screen binders in parallel and in a single XFEL experiment. » ==> The authors should use the « ligands » instead of « binders »

Typo p9: change « denisty » by « density »

REVIEWER COMMENTS

Reviewer #1 (Remarks to the Author):

The manuscript by Wranik, Kepa, Beale et al. is excellent and provides a significant step forward in using HVE injection for serial crystallography and thus opening new possibilities in time-resolved structural biology and structure-based drug discovery. It reads very well, is scientifically sound and I fully support publication in Nature communications after some corrections:

1)In the abstract the authors write “and offered a path forward for the observation of temperature-dependent structural changes.” This sounds to me as if temperature-dependent changes were probed and have been observed in the work presented here, which is not the case. I would suggest to re-phrase this, as to not confuse readers of the article.

We agree that this could be a confusing sentence for readers. We have changed the text to:

“In-built temperature control of samples facilitated optimal extrusion and supported sample stability.”

2)Line 27. The authors are talking about experiments using HVE injectors and are citing “Mehrabi, P. et al. Time-resolved crystallography reveals allosteric communication aligned with molecular breathing. Science . 365, 1167–1170 (2019).”, which is an excellent example for TR-SSX, however no HVE injection was used to obtain the results presented.

We thank the referee for spotting this error, this reference has been removed from this point in the text.

3)Line 29. only names 4th generation Synchrotrons/upgrades in Europe and not even all of them. NSLS-II is already in operation, the APS upgrade is on-going and in Europ the PETRAIV upgrade will happen only slightly after the Diamond-II upgrade.

We have added information to include the sources mentioned by the referee and thank them for their suggestion.

4)Line 31/32 this is only true, if HVE injectors are widely used in SSX. I would re-phrase this a bit.

We have rephrased this section and removed this sentence to make it a more general statement.

5)I would simply cut this sentence: “We note that in terms of sample delivery, the higher flux of the 4th generation synchrotrons will present challenges similar to those at XFELs, e.g., handling a large number of micron-size crystals.”

We have removed this sentence as suggested.

6)Line 89 -91: “A guiding aperture then ensures an accurate fit of the capillary needle of the sample reservoir into the ceramic gas aperture. The gas aperture is connected to a helium

gas supply fitted with a remotely controlled inline pressure regulator.” How is the ceramic gas aperture sealed towards the back (the exchangeable sample capillary). Would be good to elaborate a bit on that.

In order to clarify this, we have added the following text to the supplementary information. We felt that it was more appropriate here than in the main text so as not to disrupt the flow of the main text with rather detailed information.

“After the guiding aperture, the capillary is driven through a small hole with an approximate diameter of 430 μm which forms the ‘back’ of the gas aperture mount. This seals the ‘back’ of the capillary in the gas aperture mount sufficiently well so that He gas flow is preferentially delivered through the larger opening of the ceramic gas aperture around the extruded sample”

7)Line 118-120: “The loading sequence of the multi-reservoir (sample selection and movement of the chosen sample into the sample interaction region) takes approximately 60-120 seconds depending on the volume of sample loaded.” Why does this depend on the volume of sample loaded?

We have added the following text to clarify this statement:

“This dependence on sample volume is simply a result of the motor having to translate over a larger distance in order to reach the sample if a smaller volume is loaded.”

8)Line 157-158: “These data were used to determine difference maps $F_{\text{olight}}-F_{\text{odark}}$ and extrapolated maps (Supplementary Figure S6),” what kind of isomorphous difference maps are you using? q -weighted generated with Xtrapol8? Then you should mention this here and also cite Xtrapol8 already here. Same is true for all other $F_{\text{o}}-F_{\text{o}}$ maps. Please always mention if they are standard $F_{\text{o}}-F_{\text{o}}$ maps or q - or k -weighted (for example line 250 and so on).

The KR2 maps were generated using Xtrapol8. Specifically, they are q -weighted $F_{\text{o}}-F_{\text{o}}$ maps. We thank the reviewer for noticing the lack of reference to Xtrapol8 here, this has been amended in the revised manuscript and the details of maps have been included in the text for clarity.

The text has been modified as follows: “Xtrapol8 was used to calculate q -weighted difference maps $F_{\text{o}}^{\text{light}}-F_{\text{o}}^{\text{dark}}$ and corresponding extrapolated maps from these data (Supplementary Figure S6). These maps depicted the structural changes 1 μs after retinal photoactivation (Figure 2C).

We have also adjusted the caption of Figure 2 to read: “Difference electron density maps ($qF_{\text{o}}^{\text{light}}-F_{\text{o}}^{\text{dark}}$) shown at sigma level of 3.0 depicting the structural changes of and around the retinal 1 μs after photoactivation.”.

9)Line 226: should be “difference density” not “difference denisty”

We have corrected this typo.

10)Line 436-442: the information about search models is important, but could be included in the crystallographic table and omitted here.

We thank the referee for this comment. However, to our understanding, the journal prefers this information to be included in the methods section. Therefore, we have kept this part as it is in the revised manuscript.

11)Figure 1: a schematic, like in panel B, but for the whole assembly, including the interaction region would be good to understand better the set-up.

We appreciate this suggestion. When making this figure, we hoped to outline the overall components and performance of the device. Given the wide audience of this journal, we felt that more detailed descriptions of the device were best given in the supplementary information for interested readers. To that end, we feel that this information is covered sufficiently in Supplementary Figures 1-3.

12)Figure 2: what is the reason for adding a second 2Fo-Fc map at a sigma-level of 3.6? It would look better without it and gives not much additional information.

We thank the author for their comment. The figure now has a single map only for improved clarity.

13)Figure 4: “and serial” instead of “andserial” (caption line 1)

We thank the referee for spotting this error. The first line of this caption has now changed and as a result does not include this typo. It now reads “Comparison of different data analysis approaches for detecting low occupancy binding of SolQ2Br.”.

14)Figure 5:” Comparison of conventional and serial protein crystallography”. What does conventional mean here? Did you collect additional data using standard cryo-MX? If not, please remove “conventional” and re-phrase

We apologise as this error was mistakenly retained from a previous version of the text. This has now been corrected to “Comparison of different data analysis approaches for detecting low occupancy binding.”

15)Table 1: what about CC* in addition to CCI/2? Also, you mention use of Staraniso for KR2, what do the “aniso” data processing stats for lysozyme refer to? Also you mentioned you used Staraniso for TD1, however there is no “aniso” column for the TD1-datasets.

We thank the referee for this comment. The lysozyme data showed anisotropy and so were additionally processed in this way. We had accidentally omitted this from the methods section and have amended this by adding the sentence “The staraniso server was used for anisotropic correction of obtained data”. The signal to noise of the highest resolution shell is significantly improved when the data are processed in this way. TD1 was indeed analysed anistropically as well, and we thank the reviewer for spotting that the columns were missing from the table. This has now been amended.

16)Supplementary figure S2: it would be good to label all the parts. Panels A and B could also be moved to main and merged with figure 1.

We thank the referee for their suggestions. Supplementary Figure S2 has been labelled to indicate the parts within so that this is clearer. Regarding merging parts of the supplementary figures with Figure 1: We feel that the Figure 1 panel should remain an overview of the device so as not to overload the reader with too much detail. Our hope is that more interested readers are able to access further details of the device in the supplementary text. As such, we suggest keeping Figure 1 as it appears.

REVIEWER COMMENTS

Reviewer #2 (Remarks to the Author):

The paper by Wranik, Kepa, Beale et al. reports on a new multi-reservoir extruder device for serial presentation of crystalline samples at X-ray Free Electrons lasers and synchrotrons. The idea of a rotating barrel to store sample cartridges is appealing, as is the possibility of collecting data at varying temperature, e.g. to obtain insights into equilibrium dynamics. The authors also did a good job at demonstrating that their system is suited for static, time-resolved and ligand based crystallographic studies, and this referee has no doubt that their system will be used and useful by/to the (hopefully-growing) serial-crystallography community. However, it is unclear what in this paper is novel enough to warrant publications in Nature Communications. Indeed, (i) the lysozyme structure is evidently not new; (ii) the KR2 results are not new ; (iii) the binding sites on tubulin were all already known; (iv) the 'evolution' of the extruder is incremental. Maybe would the feeling of not learning much have been less present if insights into equilibrium dynamics had been by obtained by collection of data at varying temperatures, or if the pump-probe experiment had been carried out on the complex of tubulin with the photochemical affinity switch SBTub4. But this is not the case, and therefore the existence of this device is the only thing one learns from the paper.

We appreciate the above feedback and would like to address the following points:

(i) the lysozyme structure is evidently not new

The well-known lysozyme crystals were used as a benchmark to prove that the multi-injector does not hinder or reduce the data quality in any way. They were also used to demonstrate the application of the multi-injector with two different sample delivery media LCP and HEC. We strongly feel that without the well-established controls (including a membrane protein control) assessment of the performance of a new device would be very challenging and open to error.

(ii) the KR2 results are not new

As similar to the above, the KR2 sample was used to illustrate that the multi-injector permits the collection of time-resolved data and that the quality of this data is not significantly different to that collected using previously described HVE jets. We used it as a reference and openly state why this system was chosen.

(iii) the binding sites on tubulin were all already know

We agree, however, our work is one of the first systematic studies of binding sites in a pharmacologically important protein system using data recorded at an XFEL. As explained above, we strongly feel that such a demonstration is needed to confidently establish our new XFEL delivery system. Furthermore, we present the structure and binding pose of a novel photo-switch with future potential applications.

(iv) evolution of the extruder is incremental

The principle idea of sample extrusion is not new and we do not claim we have a new type of extruder. We do, however, claim that we have made a significant, not incremental, step towards full automation of viscous extruders. As explained in our work, this step required the design and in-depth evaluation of a novel instrument.

- (v) *feeling of not learning much have been less present if insights into equilibrium dynamics had been by obtained by collection of data at varying temperatures, or if the pump-probe experiment had been carried out on the complex of tubulin with the photochemical affinity switch SBTub4. But this is not the case, and therefore the existence of this device is the only thing one learns from the paper.*

We agree that studying protein dynamics in a pump-probe experiment or by varying temperature would add significant content to the manuscript. However, we strongly believe that such studies would exceed the scope of our work. Indeed, we published a separate paper on ligand-protein dynamics [Wranik et al Nature Comm 14, 903 (2023)] and our work on temperature-dependent systems is ongoing.

We also believe that the paper describes new results over and above the existence of the device. Specifically, we devoted a significant part of our work to the comprehensive analysis and discussion of data analysis methods used to find low occupancy binders. These methods can be applied in order to further improve the efficiency of experiments at XFELs. Indeed, we were able to determine the previously unknown structure and binding pose of a novel photo-switch bound to tubulin with low occupancies down to 30%.

Together with the presentation of a new sample delivery system for XFEL experiments, we feel that our work provides more than adequate learnings that will allow planning, streamlining, and analyzing XFEL experiments in a manner that goes beyond the current state-of-the art.

- (vi) *Before resubmission of the paper (to this journal or another), the authors should devote some time to proof-reading. Notably, they should verify their reference list:*
- *in some references, some authors are listed with their full name, others with their surname in full but initials for the first name, and yet others with only initials (see e.g. ref. 2)*
 - *In other references, we have the name of the website that is printed along with that of the paper (ref 3)*
 - *Other references have new words in their title (e.g. ref 6: « research papers » is appended to the title.*
 - *Other references have authors listed in an incorrect order (e.g. ref 58)*
 - *Some references are duplicated (e.g. ref 79 and 81).*
- The authors should correct for these mistakes (which are not only details) as they leave the reader with the bad impression of an hastingly written paper...*

We appreciate the comment and fully agree with the reviewer. We believe we have fixed these typos and have rectified the issues with the references.

- (vii) « With the advent of diffraction limited 4th generation synchrotrons (and dedicated beamlines) including: MAX IV (BioMAX)²⁷ and ESRF30 (ID29)²⁸ both in operation as well as Diamond-II and the SLS2.0, planned operation in 2027 and 2025, respectively, it is evident that serial crystallography methods, including HVE injectors, are becoming even more widespread. » ==> the authors should rephrase that sentence.

We appreciate the comment. This section of the text has been rephrased to the following: “At synchrotron sources, HVE injectors have also been adapted for both static^{21–23} and time-resolved serial crystallography^{24,25} for probing structural changes on the order of milliseconds. With the advent of diffraction limited 4th generation synchrotrons and beamlines either dedicated to or capable of serial crystallography measurements, it is evident that serial crystallography methods are becoming even more widespread. Examples currently in operation include MAX IV (BioMAX)²⁶, NSLS II (FMX)²⁷ and ESRF (ID29)²⁸. Further examples of planned upgrades include the APS²⁹, SLS2.0³⁰, Diamond-II³¹ and PETRA-IV³². These are expected to be operational in 2024, 2025, 2027 and 2029, respectively.”

- (viii) « We note that in terms of sample delivery, the higher flux of the 4th generation synchrotrons will present challenges similar to those at XFELs, e.g., handling a large number of micron-size crystals. » ==> the authors should explain why.

We appreciate the comment. Having reworded the text leading up to this point (as suggested by the reviewer in their previous comment) we have decided to remove this sentence for clarity.

- (ix) « In this work, we describe a new multi-reservoir device for viscous sample extrusion, which was designed to address some of the limitations of existing HVE injectors^{15,23,29} as well as to automate the sample exchange process in a serial crystallography experiment. » ==> The authors could list some of these limitations so that it is clear which of these are addressed by their new design (and which aren't).

We appreciate the comment. The following text has been added to the revised manuscript: “In this work, we describe a new multi-reservoir device for viscous sample extrusion, which was designed to address some of the limitations of existing HVE injectors^{15,23,33}, particularly the timely manual exchange of sample reservoirs. The device was also designed to automate the sample exchange process in a serial crystallography experiment in general.”

Specific developments of the system are described in detail in the Results section “Features and design of the multi-reservoir high viscosity extruder” and we feel these are more appropriately described here. These improvements include a precise read out of the amount of sample remaining, the ability to precisely align the capillary with respect to the gas aperture using motorised stages, a tip cleaning system to reduce the need for manual intervention (if jetting is poor or after a blockage) and an adjustable catcher system which again reduces the need for manual intervention. The ability to control the temperature of the reservoirs and thus the sample within is also an advantage. However, this is a feature that exists already in the HVE system described in Shimazu et al., 2019 (<https://doi.org/10.1107/S1600576719012846>).

- (x) *« These reservoirs (loaded simultaneously into the multi-reservoir extruder) were sequentially measured without disrupting the evacuated Prime chamber or beamline hutch. » ==> the authors should rephrase that sentence, notably the term « evacuated Prime chamber ».*

We appreciate the comment. We have changed the sentence to: “These reservoirs (loaded simultaneously into the multi-reservoir extruder) were sequentially measured without disrupting the low-pressure helium environment of the Prime chamber and without entering the beamline experimental hutch.”

- (xi) *« This information cannot easily be obtained by any other means^{2–4,22,60–62} both experimentally, such as time-resolved cryo-EM where time resolution is limited to tens of ms^{63–65,275} and computational such as AlphaFold⁶⁶. » ==> the authors should rephrase that sentence.*

We appreciate the comment. We have rephrased it to: “To this day TR-SFX remains the most developed and successful technique to visualise protein dynamics down to ultrafast timescales. These timescales still remain inaccessible using other methods, as discussed elsewhere^{2–4,22,64–66}, whether experimental, such as time-resolved cryo-EM (limited to ms^{67–69}), or computational, such as AlphaFold⁷⁰.

- (xii) *« This suggests that in the future, a combined approach relying on ligand cocktails and the multi-reservoir extruder could be used to efficiently and rapidly screen binders in parallel and in a single XFEL experiment. » ==> The authors should use the « ligands » instead of « binders »*

We appreciate the comment and have changed the text accordingly.

- (xiii) *Typo p9: change « denisty » by « density »*

We appreciate the comment. The typo has been corrected.

Reviewers' Comments:

Reviewer #1:

Remarks to the Author:

Thank you for your replies to my comments! They have all been addressed properly and I can fully recommend publication in Nature communications.